# DoFIT: Domain-aware Federated Instruction Tuning with Alleviated Catastrophic Forgetting

**Binqian Xu**[1], **Xiangbo Shu**[1,*], **Haiyang Mei**[2], **Zechen Bai**[2], **Basura Fernando**[3],
**Mike Zheng Shou**[2], **and Jinhui Tang**[1]

[1]Nanjing University of Science and Technology    [2]Show Lab, National University of Singapore
[3]Institute of High-Performance Computing, A*STAR
https://github.com/1xbq1/DoFIT

## Abstract

Federated Instruction Tuning (FIT) advances collaborative training on decentralized data, crucially enhancing model's capability and safeguarding data privacy. However, existing FIT methods are dedicated to handling data heterogeneity across different clients (i.e., client-aware data heterogeneity), while ignoring the variation between data from different domains (i.e., domain-aware data heterogeneity). When scarce data needs supplementation from related fields, these methods lack the ability to handle domain heterogeneity in cross-domain training. This leads to domain-information catastrophic forgetting in collaborative training and therefore makes model perform sub-optimally on the individual domain. To address this issue, we introduce **DoFIT**, a new **Do**main-aware **FIT** framework that alleviates catastrophic forgetting through two new designs. First, to reduce interference information from the other domain, DoFIT finely aggregates overlapping weights across domains on the inter-domain server side. Second, to retain more domain information, DoFIT initializes intra-domain weights by incorporating inter-domain information into a less-conflicted parameter space. Experimental results on diverse datasets consistently demonstrate that DoFIT excels in cross-domain collaborative training and exhibits significant advantages over conventional FIT methods in alleviating catastrophic forgetting. Code is available at https://github.com/1xbq1/DoFIT.

## 1 Introduction

Large Language Models (LLMs) have attracted significant attention due to their remarkable comprehension and reasoning capabilities, coupled with their vast potential for a wide array of applications [27]. This attention has spurred the development of various Parameter Efficient Fine-Tuning (PEFT) methods [13, 15, 14, 6], aimed at efficiently adapting these powerful models to specific tasks under constrained computational resources [21]. Among them, LoRA [6] stands out as one of the most popular, due to its lower number of trainable parameters and the absence of additional inference computations. While LoRA significantly alleviates the computational burden associated with tuning LLMs, substantial challenges persist at the data level [28, 31], particularly in domains involving privacy concerns, where there is a lack of high-quality for cultivating a strong model [26, 29].

Towards this issue, some Federated Instruction-Tuning (FIT) methods have been explored by combining LLM instruction-tuning using LoRA with Federated Learning (FL) in recent years [37, 35, 38, 11, 34, 4, 9, 24]. In such FIT methods, the server side coordinates multi-round training among clients without sharing data for boosting model capability and protecting data privacy. Specifically, each round consists of four steps: global LoRA downloads from the server back to the clients, local

---

*Corresponding author

38th Conference on Neural Information Processing Systems (NeurIPS 2024).

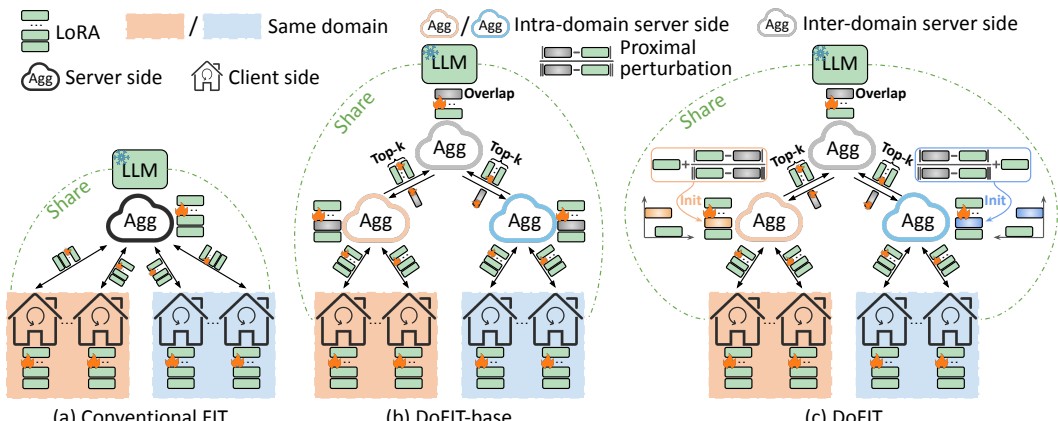

Figure 1: (a) **Conventional FIT** (with LoRA): directly expands from intra-domain to inter-domain settings. (b) **DoFIT-base** (with catastrophic forgetting): aggregates overlapping modules among the top-$k$ important modules from different domains on the inter-domain server side and completes the personalized initialization of the updating weight matrix on the intra-domain server side by assigning values to corresponding modules while keeping the rest unchanged. (c) **DoFIT** (with alleviated catastrophic forgetting): further integrates a proximal perturbation initialization strategy into the DoFIT-base for alleviating catastrophic forgetting in terms of domain information.

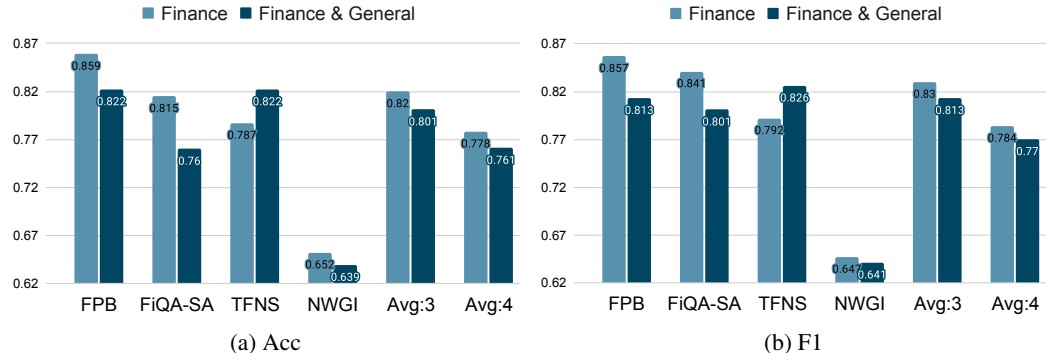

Figure 2: Performance effect of conventional FIT trained on Specific domain (i.e., Finance) and Finance&General domain. FinGPT [36] and Alpaca-GPT4 [23] are the training datasets on Finance domain and General domain, respectively. FPB [19], FiQA-SA [18], TFNS [17], and NWGI [33] are all the evaluation datasets on Finance domain. Avg:3 and Avg:4 denote the average result on the first three evaluation datasets (i.e., FPB, FiQA-SA, and TFNS) and all the evaluation datasets, respectively.

LoRA updates on the client side, local LoRA uploads from clients to the server, and global LoRA aggregates on the server side.

Currently, most FIT methods are dedicated to handling data heterogeneity between clients, i.e., client-aware data heterogeneity. When data within a specific domain is scarce and needs to be supplemented with data from other related domains to develop a powerful model, existing FIT methods treat intra- and inter-domain data heterogeneity (i.e., domain-aware data heterogeneity) equally and cannot adapt to the variation in cross-domain training, as shown in Figure 1 (a). This variation comes from more common information in the same domain and more interference information from different domains. Furthermore, as shown in Figure 2, conventional FIT with LoRA often suffers from catastrophic forgetting of domain information, and performs suboptimally on each individual domain, due to its inability to handle the domain-aware data heterogeneity. To tackle domain-aware data heterogeneity well, two main parts in conventional FIT, i.e., aggregation and initialization, are evolved in this work.

**Revisiting Aggregation in FIT.** Considering the variation in intra-domain and inter-domain data, we design a domain-aware FIT baseline (DoFIT-base) for separately aggregating intra- and inter-domain

information, as shown in Figure 1 (b). In DoFIT-base, it first normally aggregates domain-specific information on the intra-domain server side, and then aggregates overlapping domain-agnostic information at a fine granularity in the inter-domain server side. In the inter-domain server side, this fine-grained aggregation strategy aims to reduce interference from irrelevant information. This strategy mainly includes two steps. The first step is to upload only the top-$k$ important modules from each intra-domain server side. It is noticeable that that DoFIT-base still uses LoRA ($A$ or $B$ as one module), with each layer of LoRA focusing on different aspects [30]. The larger the LoRA weight is, the greater its impact on the frozen LLM becomes, and thus, it becomes more important [40]. The second step is to averagely aggregate the overlapping modules uploaded from different intra-domain server sides. This step is necessary because the top-$k$ modules uploaded by different intra-domain server sides are different. After aggregation on the inter-domain server side, each intra-domain server side obtains important and necessary module information for itself.

**Revisiting Initialization in FIT.** To retain more domain information, inspired by the orthogonal learning approach in [12], we introduce a new initialization strategy based on the proximal perturbation by projecting the modules with inter-domain information onto parameter regions least affected by intra-domain update. Specifically, as shown in Figure 1 (c), after completing the aggregation on the inter-domain server side, the newly generated modules are transmitted to the intra-domain server side. On the intra-domain server side, proximal perturbations are calculated between the new and the original modules, and then added to the original modules rather than being directly overwritten. The proximal perturbation term contains inter-domain information, while the original modules retain global intra-domain information. This less-conflicted initialization strategy can more effectively preserve domain information, ultimately mitigating catastrophic forgetting.

In summary, our contributions are: 1) the first solution to concern domain-aware data heterogeneity in collaborative training on decentralized data for the FIT paradigm; 2) a new domain-aware FIT framework that involves fine-grained inter-domain aggregation to handle domain-aware data heterogeneity; 3) a novel initialization strategy in intra-domain global LoRA to alleviate catastrophic forgetting in terms of domain information; and 4) the significant performance gain over conventional FIT and comprehensive analysis to pave the way for future explorations into more advanced FIT.

# 2 Related Works

## 2.1 Federated Instruction Tuning.

Under the condition of protecting client data privacy, Federated Instruction Tuning (FIT) enables collaboration on high-quality instruction data, facilitating the instruction-tuning of pre-trained LLMs for downstream tasks aimed at understanding diverse human intentions [37, 35, 32]. As the pioneer work, Zhang et al. [37] introduced a basic framework, which adopts LoRA for conducting client-side updating and server-side aggregation. Compared to the basic initial framework, Kuang et al. [11] provided a comprehensive framework that covers data processing, federated training, and multiple benchmarks, while also implementing various Parameter-Efficient Fine-Tuning (PEFT) methods, memory-saving operations, as well as acceleration techniques.

Since the above FIT methods neglect alignment with human values, Ye et al. [35] presented federated value alignment alongside federated instruction-tuning, ensuring both harmless and helpful outputs. To tackle data extraction attacks and limited instruction data, Zhang et al. [39] proposed to employ LLM to synthesize data, and then train local models on both synthesized and original data, as well as global models solely on synthesized data. In heterogeneous scenarios with test-time distribution shift, Yang et al. [34] proposed a personalized FIT achieved by incorporating local LoRA and shared global LoRA. To confront the challenges arising from resource and data heterogeneity, Zhang et al [38] explored pruning personalized sparse structures for clients with resource imbalances using neural architecture search.

However, the above FIT methods only consider client-aware data heterogeneity, overlooking both the intra- and inter-domain data heterogeneity when different relevant domains co-exist, namely domain-aware data heterogeneity. To this end, we introduce intra- and inter-domain server sides to deal with two types of heterogeneity differently.

## 2.2 Parameter-Efficient Fine-Tuning.

The increasing parameter size of LLMs results in expensive computational costs. However, for adapting to downstream tasks, full-parameter fine-tuning of LLMs poses challenges on hardware platforms with limited computational resources [3]. To this end, various Parameter-Efficient Fine-Tuning (PEFT) methods have been proposed by freezing the pre-trained LLMs while only fine-tuning a small number of parameters [3]. Here, we mainly introduce several representative PEFT methods, i.e., Serial Adapter [5], Prefix-tuning [13], P-Tuning [15], IA3 [14], and Low-Rank Adaptation (LoRA) [6]. For more details about PEFT methods, please refer to reference [3].

Serial Adapter [5] built two adapter modules following the self-attention and FFN layers. Each adapter module comprises a down-projection matrix, a non-linear activation function, and an up-projection matrix. Prefix-tuning [13], as soft prompt method, added trainable vectors as prefixes to both the key and value of all layers, while P-Tuning [15] integrated trainable vectors as prefixes into the initial word embedding layer. Similarly, IA3 [14] added scaling trainable vectors to key, value, and FFN activations. As one of the re-parameterization methods, LoRA [6] decomposed frozen parameters into two low-rank trainable matrices during fine-turning, and merged them with LLMs during inference without extra computational overhead. Similarly, considering the efficiency of LoRA in inference, our DoFIT adopts the PEFT method with LoRA.

# 3 Methodology

## 3.1 Preliminaries

**Low Rank Adaptation (LoRA).** Considering the limited computational resources available on the client side, it is challenging to perform full-parameter instruction tuning for LLM. Fortunately, Low Rank Adaptation (LoRA) [6], the most popular Parameter-Efficient Fine-Tuning (PEFT) method in federated setting [11], has been successfully applied in Federated Instruction Tuning (FIT) [37, 35]. During adaptation for specific tasks, given the low intrinsic dimension of pre-trained language models, LoRA assumes that the update to the pre-training weight matrices similarly exhibit a low intrinsic rank [6]. Consequently, for a frozen pre-trained weight matrix $W_0 \in \mathbb{R}^{d \times k}$, its updating weight matrix $\triangle W \in \mathbb{R}^{d \times k}$ is decomposed into low-rank trainable parameters $BA$, as follows,

$$W_0 + \triangle W = W_0 + BA \tag{1}$$

where $B \in \mathbb{R}^{d \times r}$, $A \in \mathbb{R}^{r \times k}$, and the rank $r \ll min(d, k)$. $A$ uses random Gaussian initialization and $B$ is initialized with zero.

**Conventional FIT with LoRA.** Federated Instruction Tuning (FIT) is intended to resolve the issue of inadequate high-quality instructional data for individual clients and the inability to share such data due to privacy concerns. In FIT, it usually involves a server and multiple clients, where the server achieves collaborative training of non-shared instructional data among different clients by aggregating and initializing clients' updating weight matrices. To provide a detailed description of this process, we first define the updating weight matrix $\triangle W_i^t$ in the client side, as follows,

$$\triangle W_i^t = \{\triangle W_{i,l}^t\}_{l=1}^L = \{B_{i,l}^t A_{i,l}^t\}_{l=1}^L \tag{2}$$

where $r$, $i$, $l$, and $L$ denote the $t$-th round, the $i$-th client, the $l$-th layer, and total $L$ layers, respectively. During the training process, the updating weight matrix $\triangle \bar{W}_l^{t-1}$ from the server is first downloaded for initializing the updating weight matrix $\triangle W_{i,l}^t$ on the client, as follows,

$$\{\triangle W_i^t\}_{\text{init}} = \triangle \bar{W}^{t-1}$$
$$\{\{B_{i,l}^t A_{i,l}^t\}_{l=1}^L\}_{\text{init}} = \{\bar{B}_l^{t-1} \bar{A}_l^{t-1}\}_{l=1}^L \tag{3}$$

Although only the weight initialization for the $i$-th client is indicated here, all selected clients undergo the same initialization process. After initializing, the updating weight matrices are optimized (i.e., $\{B_{i,l}^t A_{i,l}^t\} \leftarrow \{B_{i,l}^t A_{i,l}^t\}_{\text{init}}$) based on their individual data, with loss function corresponding to each client's task. Subsequently, the updating weight matrices $\{\triangle W_{i,l}^t\}_{i \in \Omega_N}$ from selected client sets $\Omega_N$ are uploaded to the server side for aggregation, generating a new updating weight matrix $\triangle \bar{W}_l^t$ on

the server side. The aggregation process is shown as below,

$$\triangle \bar{W}_l^t = \text{Agg}_{i \in \Omega_N}(\triangle W_{i,l}^t)$$
$$= (\frac{1}{N}\sum_{i \in \Omega_N} B_{i,l}^t)(\frac{1}{N}\sum_{i \in \Omega_N} A_{i,l}^t) \tag{4}$$
$$= \bar{B}_l^t \bar{A}_l^t$$

where $\Omega_N$ represents a set of randomly sampled client indices, with a total of $N$ clients. The function $\text{Agg}(\cdot)$ averages $B_{i,l}^t$ and $A_{i,l}^t$ within the selected clients' corresponding layer. $\bar{B}_l^t \bar{A}_l^t$ constitutes the new aggregated updating weight matrix $\triangle \bar{W}_l^t$.

## 3.2 Domain-aware FIT Baseline (DoFIT-base)

The intra-domain and inter-domain data heterogeneities are unequal in domain-aware data heterogeneity. Conventional FIT fails to distinguish between intra- and inter-domain data heterogeneities, as it employs the same federated architecture to handle them equally, only altering the client data to be within the same domain or across different domains, as shown in Figure 1 (a). Hence, when various clients possess datasets from other relevant domains, the results of conventional FIT may be inferior to those of the original specific domain, as shown in Figure 2.

In comparison to the intra-domain scenario, data from different domains demonstrate greater heterogeneity. To benefit from information in other relevant domains, we need to carefully design the aggregation strategy to more finely extract and aggregate the shared information between the current domain and other relevant domains. Therefore, we introduce a domain-aware FIT baseline (called DoFIT-base) that completes different aggregation strategies from coarse-grained level to fine-grained level. As shown in Figure 1 (b), DoFIT-base takes into account both the intra-domain variance and inter-domain variance, where the latter is more challenging.

Specifically, DoFIT-base contains the intra-domain server side, inter-domain server side, and several client sides. First of all, in the inter-domain server side, the updating weight matrix $\triangle \widetilde{W}^{t-1}$ is defined as follows,

$$\triangle \widetilde{W}^{t-1} = \{\widetilde{B}_l^{t-1}\widetilde{A}_l^{t-1}\}_{\in \Psi^{t-1}} \tag{5}$$

where $\Psi^{t-1}$ denotes the overlapping modules ($B$ or $A$ as one module) from different domains in round $t-1$. Noted, "overlapping" refers to both the same layer and the same decomposition components. At the beginning of the first round, $\widetilde{A}_l^{t-1}$ and $\widetilde{B}_l^{t-1}$ are initialized with random Gaussian initialization and zero, respectively.

**Download Step.** In the updating weight matrix $\triangle \bar{W}_m^{t-1}$ of intra-domain server side, the overlapping $\{\bar{B}_{m,l}^{t-1}\bar{A}_{m,l}^{t-1}\}_{\in \Psi_{t-1}}$ are initialized by $\triangle \widetilde{W}^{t-1}$, while the personalized $\{\bar{B}_{m,l}^{t-1}\bar{A}_{m,l}^{t-1}\}_{\in \Psi_{t-1}^{\complement}}$ remain unchanged, as follows,

$$\{\{\bar{B}_{m,l}^{t-1}\bar{A}_{m,l}^{t-1}\}_{\in \Psi_{t-1}}\}_{\text{init}} = \triangle \widetilde{W}^{t-1}$$
$$\triangle \bar{W}_m^{t-1} = \{\bar{B}_{m,l}^{t-1}\bar{A}_{m,l}^{t-1}\}_{l=1}^L \tag{6}$$

where $m$ denotes the $m$-th domain. $\Psi_{t-1} \cap \Psi_{t-1}^{\complement} = \varnothing$, $\Psi_{t-1} \cup \Psi_{t-1}^{\complement} = \mathbb{U}$, and $\mathbb{U} = \{\bar{B}_{m,l}^{t-1}\bar{A}_{m,l}^{t-1}\}_{l=1}^L$. Next, $\triangle \bar{W}_m^{t-1}$ initializes the updating weight matrix $\triangle W_{m,i}^t$ on the $i$-th client side with the same domain, as follows,

$$\{\triangle W_{m,i}^t\}_{\text{init}} = \triangle \bar{W}_m^{t-1}$$
$$\triangle W_{m,i}^t = \{\triangle W_{m,i,l}^t\}_{l=1}^L = \{B_{m,i,l}^t A_{m,i,l}^t\}_{l=1}^L \tag{7}$$

**Upload Step.** On the client side, $\triangle W_{m,i}^t$ is updated based on local data and specific task loss. Then, similar to conventional FIT, the updated $\triangle W_{m,i}^t$ is uploaded to the intra-domain server side for aggregation, as follows,

$$\triangle \bar{W}_{m,l}^t = \text{Agg}_{i \in \Omega_N}(\triangle W_{m,i,l}^t)$$
$$= (\frac{1}{N}\sum_{i \in \Omega_N} B_{m,i,l}^t)(\frac{1}{N}\sum_{i \in \Omega_N} A_{m,i,l}^t) \tag{8}$$
$$= \bar{B}_{m,l}^t \bar{A}_{m,l}^t$$

Similar to [40], the squared norm of the module serves as the important score for itself, determining its impact on the frozen LLM. Obviously, not all modules have the same importance score. In fact, the more important a module is, the greater its influence on the instruction tuning process. To mitigate the impact of irrelevant information from other domains, only common and important information should be captured. In the intra-domain server side, all modules in $\triangle \bar{W}_m^t$ are sorted based on their important scores, where only the top-$k$ modules are selected and uploaded to the inter-domain server side. Subsequently, on the inter-domain server side, the overlapping modules across all domains are aggregated accordingly, while the modules that do not overlap across all domains remain unchanged, as follows,

$$
\begin{aligned}
\triangle \widetilde{W}_l^t &= \mathrm{Agg}_{m \in M}(\triangle \bar{W}_{m,l}^t) \\
&= (\frac{1}{M} \sum_{m \in M} \bar{B}_{m,l}^t)(\frac{1}{M} \sum_{m \in M} \bar{A}_{m,l}^t) \\
&= \widetilde{B}_l^t \widetilde{A}_l^t
\end{aligned}
\tag{9}
$$

where $\{\bar{B}_{m,l}^t \bar{A}_{m,l}^t\}_{\in \Psi_t}$ and $\{\widetilde{B}_l^t \widetilde{A}_l^t\}_{\in \Psi_t}$. $\Psi_t$ indicates the set of overlapping modules in the $t$-th round, and $M$ denotes the number of domains.

### 3.3 Domain-aware FIT (DoFIT)

In traditional FL, the iterative training across multiple rounds often results in global information forgetting from previous rounds due to the heterogeneity nature of data on client sides. This issue persists in our DoFIT-base, which is concretely manifested as the problem of domain information forgetting.

To retain more domain information and alleviate such problem, inspired by the orthogonal learning in [12], we translate inter-domain information to the parameter space least conflicted by the updating on the intra-domain server side, thereby reducing conflicts between intra- and inter-domain information. Thus, the initialization process in Eq. 6 can be modified from directly covering $\triangle \bar{W}_m^{t-1}$ by $\triangle \widetilde{W}^{t-1}$, for adding a proximal perturbation computed from the module-wise difference between $\triangle \widetilde{W}^{t-1}$ and $\triangle \bar{W}_m^{t-1}$, as follows,

$$
\{\bar{B}_{m,l}^{t-1} \bar{A}_{m,l}^{t-1}\}_{\text{init}} = \left\{ (\bar{B}_{m,l}^{t-1} + \alpha \frac{\left| \widetilde{B}_l^{t-1} - \bar{B}_{m,l}^{t-1} \right|}{||\widetilde{B}_l^{t-1} - \bar{B}_{m,l}^{t-1}||_2})(\bar{A}_{m,l}^{t-1} + \alpha \frac{\left| \widetilde{A}_l^{t-1} - \bar{A}_{m,l}^{t-1} \right|}{||\widetilde{A}_l^{t-1} - \bar{A}_{m,l}^{t-1}||_2}) \right\}_{\in \Psi_{t-1}}
\tag{10}
$$

where $\alpha$ is the scaling factor. $\frac{|\widetilde{B}_l^{t-1} - \bar{B}_{m,l}^{t-1}|}{||\widetilde{B}_l^{t-1} - \bar{B}_{m,l}^{t-1}||_2}$ and $\frac{|\widetilde{A}_l^{t-1} - \bar{A}_{m,l}^{t-1}|}{||\widetilde{A}_l^{t-1} - \bar{A}_{m,l}^{t-1}||_2}$ denote proximal perturbation terms, mapping $\widetilde{B}_l^{t-1}$ and $\widetilde{A}_l^{t-1}$ to the parameter region least affected by $\bar{B}_{m,l}^{t-1}$ and $\bar{A}_{m,l}^{t-1}$. The overall algorithm process of DoFIT is described in the supplemental material due to space limitations.

## 4 Experiments

### 4.1 Experimental Settings

**Datasets.** We train our DoFIT on three datasets, i.e., FinGPT [36], Alpaca-GPT4 [23], and MedAlpaca [2] from the Finance (F), General (G), and Medical (M) domains, respectively. In the F domain, FinGPT is an open-source dataset for financial sentiment analysis, consisting of 77k samples. In G domain, Alpaca-GPT4 comprises 52k instances of English instruction-following data, generated by GPT-4 [1] using identical prompts as Alpaca. In M domain, MedAlpaca includes 34k question-answer pairs sourced from the Anki medical curriculum flashcards.

**Configurations.** In all experiments conducted on one NVIDIA A40, the frozen LLM used is Llama2-7B with 32 layers [27] quantized to int8. The LoRA rank and alpha are set to 32 and 64, respectively. The maximum sequence length is 512. Following the formatting instructions of Alpaca template [25], the training runs for 200 rounds, with a cosine learning rate scheduler adjusting the learning rate from $5e-5$ to $1e-6$. In each round, the selected clients are trained 10 steps by AdamW [16] optimizer. The batch size is set to 16. In FinGPT/Alpaca-GPT4/MedAlpaca training, total 10k/20k/20k samples for 50/20/20 clients, selecting 5/2/2 clients randomly per round. Similar to [35], each training dataset

Table 1: Comparing "Local", Conventional FIT ("FIT"), DoFIT-base ("Base"), and "DoFIT" on Finance (F) domain and Finance&General (F&G) domain datasets. FinGPT [36] and Alpaca-GPT4 [23] are the training datasets on F domain and G domain, respectively. FPB [19], FiQA-SA [18], TFNS [17], and NWGI [33] are the evaluation datasets on F domain. Avg:3 and Avg:4 denote the average result on the first three evaluation datasets (i.e., FPB, FiQA-SA, and TFNS) and all evaluation datasets, respectively. ↑ refers to the performance improvement compared to the alternative marked with the same color (i.e., using the same LoRA configuration) on F domain. ↓ denotes performance degradation, oppositely.

| Domain | Method | FPB | | FiQA-SA | | TFNS | | NWGI | | Avg:3 | | Avg:4 | |
|---|---|---|---|---|---|---|---|---|---|---|---|---|---|
| | | Acc | F1 | Acc | F1 | Acc | F1 | Acc | F1 | Acc | F1 | Acc | F1 |
| F | GPT-3.5 | 0.781 | 0.781 | 0.662 | 0.730 | 0.731 | 0.736 | - | - | 0.725 | 0.749 | - | - |
| | GPT-4 | 0.834 | 0.833 | 0.545 | 0.630 | 0.813 | 0.808 | - | - | 0.731 | 0.757 | - | - |
| | Local | 0.770 | 0.760 | 0.655 | 0.719 | 0.742 | 0.747 | 0.629 | 0.624 | 0.722 | 0.742 | 0.699 | 0.713 |
| | $FIT_{32qv}$ | 0.859 | 0.857 | **0.815** | **0.841** | 0.787 | 0.792 | **0.652** | **0.647** | 0.820 | 0.830 | 0.778 | 0.784 |
| | $FIT_{16qvd}$ | 0.850 | 0.846 | 0.818 | 0.842 | 0.823 | 0.823 | 0.646 | 0.643 | 0.830 | 0.837 | 0.784 | 0.789 |
| | $FIT_{32d}$ | **0.860** | **0.857** | 0.807 | 0.836 | **0.824** | **0.825** | 0.635 | 0.635 | **0.830** | **0.839** | **0.782** | **0.788** |
| F&G | $FIT_{32qv}$ | 0.822 | 0.813 | 0.760 | 0.801 | 0.822 | 0.826 | 0.639 | 0.641 | 0.801↓ | 0.813↓ | 0.761↓ | 0.770↓ |
| | $Base_{top10}$ | 0.859 | 0.855 | 0.778 | 0.815 | 0.810 | 0.811 | 0.637 | 0.638 | 0.816 | 0.827 | 0.771 | 0.780 |
| | $Base_{top15}$ | 0.862 | 0.860 | 0.804 | 0.834 | 0.857 | 0.858 | 0.639 | 0.639 | 0.841↑ | 0.851↑ | 0.791↑ | 0.798↑ |
| | $Base_{top20}$ | 0.859 | 0.855 | 0.775 | 0.815 | 0.866 | 0.864 | 0.632 | 0.634 | 0.833 | 0.845 | 0.783 | 0.792 |
| | $DoFIT_{\alpha=0.5}$ | **0.865** | **0.861** | 0.815 | 0.842 | 0.864 | 0.864 | **0.645** | **0.644** | 0.848 | 0.856 | 0.797 | 0.803 |
| | $DoFIT_{\alpha=1.0}$ | 0.861 | 0.858 | **0.818** | **0.847** | **0.869** | **0.869** | 0.641 | 0.640 | **0.849**↑ | **0.858**↑ | **0.797**↑ | **0.804**↑ |
| | $DoFIT_{\alpha=1.5}$ | 0.859 | 0.855 | 0.815 | 0.845 | 0.822 | 0.825 | 0.642 | 0.641 | 0.832 | 0.842 | 0.785 | 0.792 |

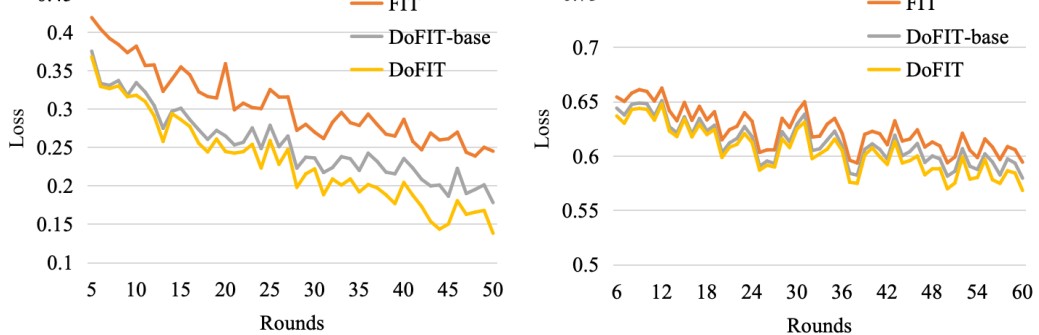

Figure 3: Loss curves for different methods, i.e., FIT, DoFIT-base, and DoFIT, in F&G (left) and M&G (right) domains, respectively.

is randomly shuffled and then evenly divided among the clients. The training datasets consist of either single-domain datasets or dual-domain datasets. The testing process is carried out on the evaluation datasets of a single domain, after merging the updating weight matrix on the intra-domain server side with the frozen LLM.

**Comparison Methods.** To validate the effectiveness of the proposed DoFIT-base (called "Base" for conciseness) and "DoFIT" in addressing intra- and inter-domain data heterogeneity, it is essential to compare them with "Local" and "FIT". Here, "Local" refers to training independently based solely on data from a single client. "FIT" [35] refers to the collaborative training of different client data based on FedAvg [20], treating client data from different domains equally. $FIT_{32qv}$ uses all layers' LoRA[Q,V] (decomposed from Q and V components of self-attention). $FIT_{16qvd}$ uses half of all layers' LoRA[Q,V,D] (decomposed from the Q, V components of self-attention and Down linear layer of MLP). $FIT_{32d}$ uses all layers' LoRA[D] (decomposed from the Down linear layer in MLP). Noted, the aforementioned Conventional FIT is $FIT_{32qv}$ [35]. For fairness, we also use FedAvg in the collaboration training from intra-domain data. In Specific and General domains, "$Base_{top10}$" refers to uploading the top-10 important modules from the intra-domain server side to the inter-domain one, based on the best baseline in the Specific domain. The rest of "$Base_{top*}$" has a similar definition. "DoFIT" modifies the initialization of the best "Base" by incorporating the proximal perturbation initialization. "$DoFIT_{\alpha=1.0}$" means that the scaling factor in DoFIT is set to 1.0.

Table 2: Comparing "Local", Conventional FIT ("FIT"), DoFIT-base ("Base"), and "DoFIT" on Medical domain (M), and combined Medical&General domain (M&G). MedAlpaca [2], and Alpaca-GPT4 [23] are the training datasets on M domain, and G domain, respectively. MedQA [10], and MedMCQA [22] are the evaluation datasets on M domain. ↑ refers to the performance improvement compared to the alternative marked with the same color (i.e., using the same LoRA configuration) on M domain.

| Domain | Method | MedQA | MedMCQA | Avg |
|---|---|---|---|---|
| M | Local | 0.141 | 0.204 | 0.173 |
| | $FIT_{32qv}$ | 0.167 | **0.216** | **0.192** |
| | $FIT_{16qvd}$ | **0.179** | 0.200 | 0.190 |
| | $FIT_{32d}$ | 0.158 | 0.199 | 0.179 |
| M&G | $FIT_{32qv}$ | $0.174\uparrow_{0.007}$ | $0.217\uparrow_{0.001}$ | $0.196\uparrow_{0.004}$ |
| | $Base_{top25}$ | 0.182 | 0.207 | 0.195 |
| | $Base_{top30}$ | $0.192\uparrow_{0.025}$ | $0.218\uparrow_{0.002}$ | $0.205\uparrow_{0.013}$ |
| | $DoFIT_{\alpha=1.1}$ | 0.253 | 0.252 | 0.252 |
| | $DoFIT_{\alpha=1.2}$ | $\mathbf{0.261}\uparrow_{0.094}$ | $\mathbf{0.255}\uparrow_{0.039}$ | $\mathbf{0.258}\uparrow_{0.066}$ |
| | $DoFIT_{\alpha=1.3}$ | 0.256 | 0.247 | 0.251 |

## 4.2 Performance Evaluation

**Comparison on F Domain and F&G Domain.** After training on FinGPT for Finance (F) domain or FinGPT/Alpaca-GPT4 for Finance&General (F&G) domain, we test the models at round 50 on the evaluation datasets for F domain, i.e., FPB [19], FiQA-SA [18], TFNS [17], and NWGI [33], as shown in Table 1. Compared to the independently trained "Local", the collaboratively trained "FIT", "$Base_{16qvd}$" and "$Base_{32d}$" via FL perform better. This indicates that training can benefit from other clients' data. Expanding from F to F&G domain, the performance of "FIT" even declines, while our "$Base_{top15}$" improves and surpasses that of FIT. This validates the effectiveness of our DoFIT-base for addressing domain-aware data heterogeneity. In F&G domain, "DoFIT" further improves performance compared to "$Base_{top15}$". This indirectly validates that the proposed DoFIT retains more inter-domain information, enhancing overall performance.

**Comparison on M Domain and M&G Domain.** After training on MedAlpaca for Medical (M) domain or MedAlpaca/Alpaca-GPT4 for Medical&General (M&G) domain, we test the models at round 60 on the evaluation datasets for Medical (M) domain, i.e., MedQA [10], and MedMCQA [22]. As shown in Table 2, we can see that: 1) "FIT", "$Base_{16qvd}$", and "$Base_{32d}$" consistently outperform "Local", indicating that collaborative training can enhance model's capability; and 2) on M&G domains, "$Base_{top30}$" and "DoFIT" exceed the performance of FIT, especially "DoFIT" with the proximal perturbation initialization strategy, proving the effectiveness of this strategy.

**Loss Curves of Different Methods and Hyper-parameters.** As shown in the loss curves of Figure 3, compared to "DoFIT-base", "DoFIT" consistently shows faster convergence and lower losses, as does "DoFIT-base" compared to "FIT". Noted, the losses of different methods are similar in the first rounds, while a gap emerges as the number of rounds increases. Figure 4 and Figure 5 also show the loss curves with different values of Top-$k$ and $\alpha$. we can find that the losses are insensitive to the values of these parameters to some extent.

**Comparison of Parameter Size.** Table 3 further shows the number of parameters per round on FIT, the best-performing "$Base_{top15}$"/"$Base_{top30}$", and "DoFIT" in F&G and M&G domains. Compared to FIT, DoFIT adds slight communication parameters between intra- and inter-domain server sides (indicated by S-Comm.), with little impact on well-resourced server sides (indicated by S-Comm.). Compared with "$FIT_{32qv}$", either $Base_{top15}$ or DoFIT requires fewer trainable parameters in the client side, as well as fewer communication parameters between the client side and the intra-domain server side.

## 5 Conclusion and Future Work

In this work, we introduced a novel Domain-aware Federated Instruction Tuning (DoFIT) framework towards collaborative training on more datasets in relevant domains for boosting the performance of

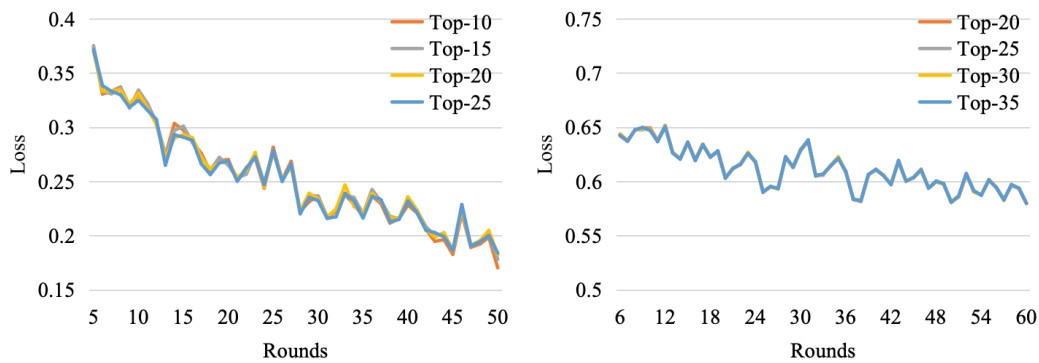

Figure 4: Loss curves for values of Top-$k$ on F&G (left) and M&G (right) domains, respectively.

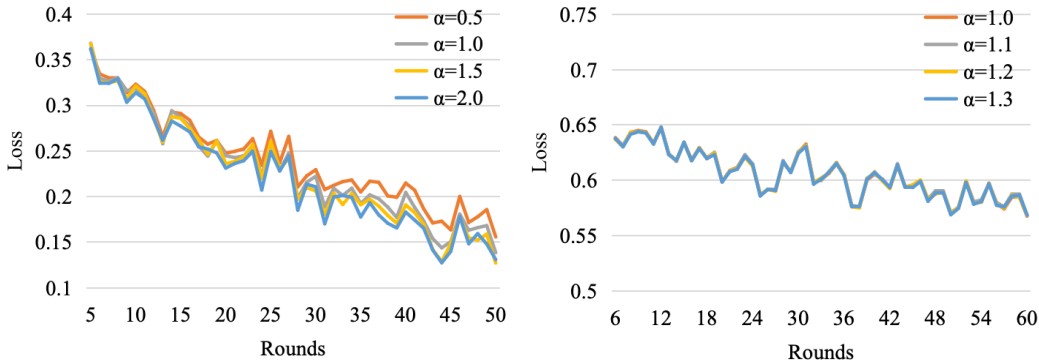

Figure 5: Loss curves for values of $\alpha$ on F&G (left) and M&G (right) domains.

Table 3: The number of parameters per round in training. "Frozen" denotes the parameter size of LLM. "Trainable" denotes the parameter size of the updating weight matrix in client side. "Comm." denotes the communication parameters between client side and (intra-domain) server side. "S-Comm." denotes the communication parameters between intra-domain server side and inter-domain server side. $32qv$ and $32d$ denote LoRA[Q,V] and LoRA[D], respectively. F&G and M&G denote Finance&General domain, and Medical&General domain, respectively.

| Domain | Method | Frozen | Trainable | Comm. | S-Comm. |
|---|---|---|---|---|---|
| F&G | FIT$_{32qv}$ | 6738M | 4.194M | 4.194M | 0M |
| | (Base$_{top15}$ / DoFIT)$_{32d}$ | 6738M | 4.021M | 4.021M | 0.942M |
| M&G | FIT$_{32qv}$ | 6738M | 4.194M | 4.194M | 0M |
| | (Base$_{top30}$ / DoFIT)$_{32qv}$ | 6738M | 4.194M | 4.194M | 0.983M |

individual domains. To alleviate the catastrophic forgetting problem caused by domain-aware data heterogeneity, our DoFIT offers two main insights in terms of aggregation and initialization. For aggregation, we first normally aggregate domain-specific information on the intra-domain server side, and then aggregate overlapping domain-agnostic information on the inter-domain server side, excluding the interference information. For initialization, we add a proximal perturbation from inter-domain information to the original modules, rather than directly overwritten them. Comprehensive experimental results on Finance, Medical, and General domains demonstrate the effectiveness of the proposed DoFIT method, compared to conventional FIT.

**Limitations.** In our experiments, we have well demonstrated that the proposed DoFIT can facilitate collaborative training on decentralized data across one specific (i.e., Finance domain, or Medical domain) domain and the General domain, significantly enhancing performance within each individual domain. DoFIT is the first attempt to concern domain-aware data heterogeneity, and keeps the FIT-like optimization strategy since it only requires the least modification to the original FIT architecture. Such succinct modification seamlessly incorporates DoFIT into the FIT family for convenient reproduction and implementation. Exploring more related domains instead of limiting to the General domain

for enhancing a specific field, and verifying DoFIT's capability to handle multiple (more than two) domains, especially when substantial domain-aware data heterogeneity exists, along with the new optimizations, will be the focus of future research.

## Acknowledgments and Disclosure of Funding

The work is supported by the National Natural Science Foundation of China (Grant No.62222207, 62072245, and 61925204), the Natural Science Foundation of Jiangsu Province (Grant No. BK20211520), the National Research Foundation Singapore and DSO National Laboratories under the AI Singapore Programme (Award Number: AISG2-RP-2020-016), and the China Scholarship Council program. Mike Shou does not receive any funding for this work.

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

# A Appendix / supplemental material

## A.1 Algorithm

---

**Algorithm 1** The training process of DoFIT for two domains

---

**Input:** $\triangle\widetilde{W}^0/\{\triangle\bar{W}_m^0\}_{m=1}^M$: Initial updating weight matrix in the inter-domain server side / intra-domain server sides, $T$ rounds, $N$: Random sample number of clients, $M$: Total number of domains, $e$: The number of epochs in the client side, top-$k$ important modules.

**Output:** $\{\triangle\bar{W}_m^T\}_{m=1}^M$
1: **for** $t = 1,\cdots,T$ **do**
2:    $\Omega_N \leftarrow$ A set of randomly sampled client indices.
3:    **for** $m = 1,\cdots,M$ **do**
4:      **if** t > 1 **then**
5:        **Intra-domain initialization**
6:        Add a proximal perturbation on Eq. 10
7:      **end if**
8:      **for** each $i$ in $\Omega_N$ **do**
9:        $\{\triangle W_{m,i}^t\}_{\text{init}} = \triangle\bar{W}_m^0$
10:       Conduct $e$ epochs of local instruction-tuning for $\triangle W_{m,i}^t$.
11:      **end for**
12:      **Intra-domain Aggregation**
13:      $\triangle\bar{W}_m^t = \text{Agg}_{i\in\Omega_N}(\triangle W_{m,i}^t)$ on Eq. 8
14:      Compute the important score of each module for $\triangle\bar{W}_m^t$.
15:      Upload top-$k$ modules for $\triangle\bar{W}_m^t$ to the inter-domain server side.
16:    **end for**
17:    **Inter-domain Aggregation**
18:    Compute the set of overlapping modules from different domains: $\Psi_t$
19:    $\triangle\widetilde{W}_l^t = \text{Agg}_{m\in M}(\triangle\bar{W}_{m,l}^t) \quad \{\bar{B}_{m,l}^t\bar{A}_{m,l}^t\}_{\in\Psi_t}$ on Eq. 9
20: **end for**

---

## A.2 Comparison with Existing Federated Domain Adaptation Works

Federated domain adaptation for LLMs is crucial, but no related methods currently exist. Applying existing federated domain adaptation methods like FedGP [8] directly to LLMs yields suboptimal results, as shown in Table 4. Where FedGP/FedGP-g refer to the projection of each client's LoRA/global LoRA weights in the source domain onto the global LoRA weights in the target domain. FPL [7] clusters prototypes from different domains into unbiased prototypes for general domain shift. However, existing federated domain adaptation methods [7, 8] for this task still merge more redundant and noisy parameters to LLMs, affecting domain fine-tuning performance. Overall, as shown in Table 1 and Table 4, our method outperforms traditional FIT and general federated domain adaptation methods.

Table 4: Comparison with existing federated domain adaptation works.

| Method | FPB | | FiQA-SA | | TFNS | | NWGI | | Avg:3 | | Avg:4 | |
|---|---|---|---|---|---|---|---|---|---|---|---|---|
| | Acc | F1 | Acc | F1 | Acc | F1 | Acc | F1 | Acc | F1 | Acc | F1 |
| FedGP | 0.837 | 0.829 | 0.760 | 0.806 | 0.789 | 0.786 | 0.625 | 0.626 | 0.795 | 0.807 | 0.753 | 0.762 |
| FedGP-g | 0.836 | 0.830 | 0.680 | 0.744 | 0.700 | 0.710 | 0.627 | 0.629 | 0.739 | 0.761 | 0.711 | 0.728 |
| DoFIT$_{\alpha=1.0}$ | 0.861 | 0.858 | 0.818 | 0.847 | 0.869 | 0.869 | 0.641 | 0.640 | 0.849 | 0.858 | 0.797 | 0.804 |

## A.3 Performance on the Gradient and Singular Value Spectrum

We add two new criteria—gradient and singular value—to assess module importance in LoRA, as follows,

Table 5: Performance on the gradient and singular value spectrum.

| Criteria | FPB | | FiQA-SA | | TFNS | | NWGI | | Avg:3 | | Avg:4 | |
|---|---|---|---|---|---|---|---|---|---|---|---|---|
| | Acc | F1 | Acc | F1 | Acc | F1 | Acc | F1 | Acc | F1 | Acc | F1 |
| DoFIT$_{\alpha=1.0}$ | 0.861 | 0.858 | 0.818 | 0.847 | 0.869 | 0.869 | 0.641 | 0.640 | 0.849 | 0.858 | 0.797 | 0.804 |
| A-grad-top15 | 0.866 | 0.864 | 0.833 | 0.852 | 0.867 | 0.867 | 0.640 | 0.639 | 0.855 | 0.861 | 0.802 | 0.806 |
| A-svd-top15 | 0.858 | 0.855 | 0.829 | 0.856 | 0.828 | 0.829 | 0.642 | 0.641 | 0.838 | 0.847 | 0.789 | 0.795 |
| B-grad-top10 | 0.823 | 0.813 | 0.789 | 0.827 | 0.802 | 0.806 | 0.633 | 0.633 | 0.805 | 0.815 | 0.762 | 0.770 |
| B-grad-top15 | 0.833 | 0.829 | 0.840 | 0.855 | 0.681 | 0.693 | 0.630 | 0.627 | 0.785 | 0.792 | 0.746 | 0.751 |
| B-grad-top20 | 0.516 | 0.480 | 0.185 | 0.197 | 0.501 | 0.500 | 0.404 | 0.369 | 0.401 | 0.392 | 0.402 | 0.387 |
| B-svd-top10 | 0.856 | 0.854 | 0.844 | 0.854 | 0.732 | 0.740 | 0.638 | 0.627 | 0.811 | 0.816 | 0.768 | 0.769 |
| B-svd-top15 | 0.821 | 0.819 | 0.793 | 0.824 | 0.621 | 0.626 | 0.644 | 0.640 | 0.745 | 0.756 | 0.720 | 0.727 |
| B-svd-top20 | 0.417 | 0.306 | 0.811 | 0.794 | 0.371 | 0.302 | 0.552 | 0.457 | 0.533 | 0.467 | 0.538 | 0.540 |

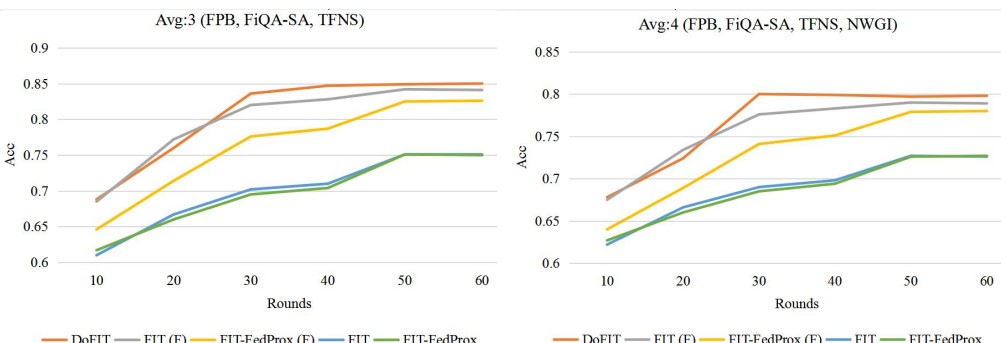

Figure 6: Comparison of average accuracy on different rounds

Using new criteria to sort modules from largest to smallest within a single domain, and select the top-k modules, like DoFIT, **Gradient:** Uses the square norm of the gradients of LoRA modules as the importance score (A-grad-top15). **Singular Value:** Uses the sum of the singular values of LoRA modules as the importance score (A-svd-top15). As shown in Table 5, the importance scores based on the gradient square norm and the sum of singular values are comparable to the module importance scores calculated using the square norm of LoRA weights in DoFIT.

From a domain distribution-aware perspective, aggregate the top-k modules with smaller domain gaps, **Gradient:** Uses the mean absolute difference of LoRA module gradients across different domains to reflect domain heterogeneity gaps (B-grad-top*). **Singular Value:** Uses the L2 norm of the differences in the singular value spectrum of LoRA modules across different domains to reflect domain heterogeneity gaps (B-svd-top*). As shown in Table 5, using gradient or singular value to aggregate modules with smaller domain heterogeneity shows more sensitivity to the top-k hyperparameter. Compared to our DoFIT, this approach performs worse. This may be because aggregating modules with smaller domain heterogeneity can still introduce redundant and noisy modules, which can degrade overall performance when merged into the LLMs.

Overall, focusing on domain-specific key parameters and removing redundancies improves performance with LLMs. DoFIT's square norm method for weights is comparable to gradient and singular value spectrum methods but is more intuitive and reproducible.

## A.4 Federated Settings Experiments

**Complexity Analysis:** As shown in Table 3, our DoFIT has the same space complexity as the traditional FIT on the client side, without any additional memory cost, but introduces a slight memory cost (S-comm.) on the inter-domain server side. In terms of time complexity, our DoFIT is identical to the traditional FIT on the client side, with only a slight computational overhead for module importance ranking on the intra-domain server side. Assuming the number of selected clients in the same domain is k, and each client includes 32d LoRA (64 modules), the sorting time complexity is

Table 6: Average accuracy on FPB, FiQA-SA, TFNS, NWGI

| Clients | Acc | F1 | Clients | Acc | F1 | Clients | Acc | F1 |
|---------|-----|-----|---------|-----|-----|---------|-----|-----|
| 50(5) & 20(2) | 0.797 | 0.804 | 50(10) & 20(2) | 0.800 | 0.806 | 50(15) & 20(2) | 0.784 | 0.792 |
| 50(20) & 20(2) | 0.783 | 0.792 | 50(5) & 20(4) | 0.786 | 0.793 | 50(5) & 20(6) | 0.794 | 0.799 |
| 50(5) & 20(8) | 0.796 | 0.802 | 50(5) & 20(10) | 0.791 | 0.797 | 25(5) & 20(2) | 0.788 | 0.794 |
| 40(5) & 20(2) | 0.748 | 0.757 | 60(5) & 20(2) | 0.791 | 0.799 | 75(5) & 20(2) | 0.791 | 0.798 |
| 50(5) & 10(2) | 0.799 | 0.804 | 50(5) & 30(2) | 0.790 | 0.797 | 50(5) & 40(2) | 0.789 | 0.793 |

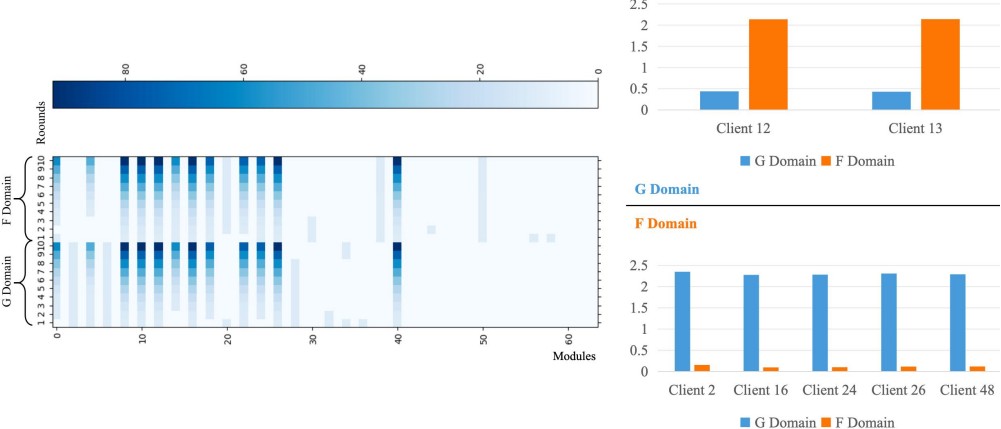

Figure 7: Modules important scores (left) and singular value spectrum (right) on F and G domains

$k \times 64 \times \log(64)$. In the financial domain, $k = 5$; in the general domain, $k = 2$; and in the medical domain, $k = 2$. The entire experiment ran on an NVIDIA A40 GPU for five and a half hours.

**Convergence Results:** As shown in Figure 6, our DoFIT demonstrates faster and more stable convergence compared to FIT using FedAvg and FedProx as the FL framework in both single-domain and dual-domain scenarios.

**Client Numbers:** 50(5) & 20(2) indicate that in the financial domain, there are 50 clients in total, with 5 clients randomly selected for training and uploading each round. In the general domain, there are 20 clients in total, with 2 clients randomly selected for training and uploading each round. As shown in Table 6, varying the total number of clients or the number of selected clients does not cause significant fluctuations in the results, demonstrating that the proposed DoFIT is robust to the number of clients.

### A.5  Domain Heterogeneity

The importance of modules in LoRA varies across different domains, indirectly reflecting domain heterogeneity. As shown in Figure 7 (left), the top-15 important modules in domains F and G are not completely the same. As training progresses, the weights of the same modules become more reinforced.

We also further compute the L2 norm of the difference in the singular value spectrum between each client's LoRA and the global LoRA for the same domain and different domains. As shown in Figure 7 (right), this visualization reflects smaller intra-domain data heterogeneity and greater inter-domain data heterogeneity.

