# OpenReview forum: "DoFIT: Domain-aware Federated Instruction Tuning with Alleviated Catastrophic Forgetting"
_NeurIPS.cc/2024/Conference — NeurIPS 2024 poster_

### Official Review · Reviewer_DeWw · 2024-06-25

**Soundness:** 3
**Presentation:** 3
**Contribution:** 3
**Rating:** 7
**Confidence:** 5

**Summary:**

This paper first introduces a new domain-aware FIT baseline called DoFIT-base. DoFIT-base aggregates domain-specific information on the intra-domain server side and domain-agnostic information on the inter-domain server side to reduce interference information from the other domains. By incorporating inter-domain information into a less-conflicted parameter space, this paper formally introduce q new DoFIT framework that retains more domain information. Since DoFIT can handle domain-aware data heterogeneity, the catastrophic forgetting problem in cross-domain training is effectively alleviated. The authors show cross-domain training results in a federated LLM trained on FinGPT, Alpaca-GPT4, and MedAlpaca, demonstrating that DoFIT performs better than conventional FIT methods.

**Strengths:**

**(1)** The idea of aggregating domain-specific/domain-agnostic information and initializing in less-conflicted parameter space is compelling and original for addressing domain-aware data heterogeneity.

**(2)** The significant performance gain over conventional FIT, along with comprehensive analysis, paves the way for future explorations into more advanced FIT.

**(3)**  Considering FIT in multi-domains in the FL scene is interesting, which can give some insights in the related filed.

**Weaknesses:**

**(1)** Why did you choose FIT_{32qv}, FIT_{16qvd}, and FIT_{32d} as comparison methods in a single domain? Would it be better to add more LoRA modules? What is the basis for choosing LoRA[D]?

**(2)** Since more servers are added to handle cases with different heterogeneity separately, additional S-comm. increases the burden of communication and security risks.

**(3)** There are many parameters in DoFIT, and as shown in Table 1 and Table 2, the parameters are different across different datasets. This variability seems to make it difficult to generalize DoFIT to other datasets.

**(4)** Does Base_{top30} in Table 2 work well, or does Base_{top35} work better?

**(5)** It is still unclear whether LoRA on the client, intra-domain server, or inter-domain server is merged with LLM for inference.

**Questions:**

Please see the comments above in the weaknesses.

**Limitations:**

The paper could benefit from more discussion on the limitations of the proposed method.

---

> ### Author Rebuttal · Authors · 2024-08-06
>
> > Weaknesses:
>
> 1: Thank you. FIT_{32qv} is the configuration of the original FIT. We experimented with FIT_{16qvd} and FIT_{32d} to explore the impact of different LoRA configurations on performance in the current domain without increasing computational and communication costs. Adding more LoRA modules would increase these costs. The choice of LoRA[D] is primarily based on the experimental results shown in Table 1 of the original paper.
>
> 2: Thanks. Compared to FIT, DoFIT adds slight communication parameters between intra- and inter-domain server sides (indicated by S-Comm.), with little impact on well-resourced server sides (indicated by S-Comm.).
>
> 3: In our DoFIT, we only introduced two hyperparameters: top-k and α. As shown in Tables 1 and 2, and Figures 4 and 5 of the original paper, the performance of DoFIT varies little with different top-k and α values. Therefore, even when generalizing to new domain data, extensive adjustment of these two hyperparameters is not necessary. The proposed DoFIT demonstrates a certain robustness to these hyperparameters.
>
> 4: Thank you. The performance of Base_{top30} in Table 2 is better.
>
> 5: The final test involves first merging the LLMs with global LoRA on the intra-domain server side, and then testing on the current domain dataset.

---

> > ### Comment · Reviewer_DeWw · 2024-08-09
> > **Response to author’s rebuttal**
> >
> > Thank you for your response. The author indeed conducted thorough ablation experiments on the LoRA parameters in the original paper and explained in the reply that more parameters would increase computational and communication costs. Additionally, the experiments on parameters k and α demonstrate a certain robustness, which suggests they might generalize more easily to other domains. This response addresses my main concerns to some extent, so I agree with raising the score.

---

### Official Review · Reviewer_zbxm · 2024-07-05

**Soundness:** 2
**Presentation:** 2
**Contribution:** 2
**Rating:** 4
**Confidence:** 4

**Summary:**

This paper proposes Federated Instruction Tuning, aimed at enhancing model's capability and data privacy. The main points lie in enhancing the model of data heterogeneity across dfifferent clients.  Specifically, the paper introduces DoFIT, a domain-aware FIT framework, aimed at alleviating the catastrophic forgetting through aggregating overlaping across domains, and incorporating inter-domain information into a less-conflict parameter space to reduce interference information from other domain. The proposed method is evaluated on diverse datasets.

**Strengths:**

1) The topic is important in federated learning .
2) The paper is easy to follow;

**Weaknesses:**

1. The novelty is limited. There are already many works in federated domain adaptation that consider the differences between domains [1][2].It seems the authors only considers large models (model differences) and uses LoRA for optimization to analyze domain heterogeneity. This approach seems to simply apply domain heterogeneity to federated learning with large models.
2. What distinguishes this work from other Federated Domain Adaptation (FDA) methods that also address domain heterogeneity?
3. I would prefer to see the authors use a more innovative criterion when considering module importance scores, rather than directly using existing work [31]. Specifically:1) Compare with other criteria, such as sorting by gradient, data consistency, etc. 2) I suggest the authors consider module importance from a domain distribution-aware perspective. Starting from domain distribution would better reflect domain heterogeneity.
4. The experimental validation is insufficient as it is entirely based on LoRA settings. Important federated settings, such as complexity, convergence, and the impact of the number of clients on the results, have not been considered.
5. An analysis of domain-heterogeneity is necessary. For example, what aspects of the model can reflect domain-heterogeneity? How can the A and B matrices in LoRA be analyzed to reflect intra-domain and inter-domain heterogeneity (e.g., using matrix singular values)
6. The algorithm's complexity needs to be analyzed, including the additional computational overhead introduced by the module importance score sorting.


[1]	Jiang, Enyi, Yibo Jacky Zhang, and Oluwasanmi Koyejo. "Federated domain adaptation via gradient projection." arXiv e-prints (2023): arXiv-2302.
[2]	Huang, Wenke, et al. "Rethinking federated learning with domain shift: A prototype view." 2023 IEEE/CVF Conference on Computer Vision and Pattern Recognition (CVPR). IEEE, 2023.

**Questions:**

1. What is the differences between the proposed method with those without large models?  The indroduction of LoRA may be a engineering problem not an technological contribution;
2. The sort critera is a little simple, what is the performance gain when used other critera?
3. The paper should include complexity, convergence, and the impact of the number of clients on the results under federated setting;

**Limitations:**

See Weaknesses.

---

> ### Author Rebuttal · Authors · 2024-08-06
>
> > Weaknesses:
>
> **1, 2:** Federated domain adaptation for LLMs is crucial, but no related methods currently exist. Applying existing federated domain adaptation methods like FedGP[1] directly to LLMs yields suboptimal results, as shown in Table 1 of the PDF. Where FedGP/FedGP-g refer to the projection of each client's LoRA/global LoRA weights in the source domain onto the global LoRA weights in the target domain. FPL[2] clusters prototypes from different domains into unbiased prototypes for general domain shift. However, existing federated domain adaptation methods[1,2] for this task still merge more redundant and noisy parameters to LLMs, affecting domain fine-tuning performance.
>
> In this work, we make the first attempt to address the LLM-oriented federated domain adaptation problem. We observe that not all LoRA parameters are useful for the current domain, where some are redundant or noisy. Aggregating all parameters can introduce irrelevant ones from other domains, weakening the influence of important parameters. The fine-tuning performance of LLM is sensitive to this. To address it, we present two innovations.  First, we sort LoRA parameters by weight and retain only the top-k to avoid the inclusion of irrelevant ones. Second, to mitigate catastrophic forgetting, we map the aggregated weights to a less conflicting parameter space. Experiments validate the effectiveness of these innovations.
>
> Overall, as shown in Table 1 of the original paper and Table 1 of the PDF, our method outperforms traditional FIT and general federated domain adaptation methods.
>
> ---
>
> **3:** Thanks. We add two new criteria—gradient and singular value—to assess module importance in LoRA. Since LoRA's data consistency is uniform, module importance scores are identical, preventing individual ranking.
>
> 1）Using new criteria to sort modules from largest to smallest within a single domain, and select the top-k modules, like DoFIT,
>
> *Gradient*: Uses the square norm of the gradients of LoRA modules as the importance score (A-grad-top15).
>
> *Singular Value*: Uses the sum of the singular values of LoRA modules as the importance score (A-svd-top15).
>
> As shown in Table 2 of the PDF, the importance scores based on the gradient square norm and the sum of singular values are comparable to the module importance scores calculated using the square norm of LoRA weights in DoFIT.
>
> 2）From a domain distribution-aware perspective, aggregate the top-k modules with smaller domain gaps,
>
> *Gradient*: Uses the mean absolute difference of LoRA module gradients across different domains to reflect domain heterogeneity gaps (B-grad-top*).
>
> *Singular Value*: Uses the L2 norm of the differences in the singular value spectrum of LoRA modules across different domains to reflect domain heterogeneity gaps (B-svd-top*).
>
> As shown in Table 2 of the PDF, using gradient or singular value to aggregate modules with smaller domain heterogeneity shows more sensitivity to the top-k hyperparameter. Compared to our DoFIT, this approach performs worse. This may be because aggregating modules with smaller domain heterogeneity can still introduce redundant and noisy modules, which can degrade overall performance when merged into the LLMs.
>
> Overall, focusing on domain-specific key parameters and removing redundancies improves performance with LLMs. DoFIT’s square norm method for weights is comparable to gradient and singular value spectrum methods but is more intuitive and reproducible.
>
> ---
>
> **4, 6:** Thanks. We conducted federated settings experiments (complexity, convergence, and client numbers) but omitted them due to space constraints and minimal impact. The paper focuses on new issues and configurations in federated domain adaptation for LLMs.
>
> *Complexity Analysis*: As shown in Table 3 of the original paper, our DoFIT has the same space complexity as the traditional FIT on the client side, without any additional memory cost, but introduces a slight memory cost (S-comm.) on the inter-domain server side. In terms of time complexity, our DoFIT is identical to the traditional FIT on the client side, with only a slight computational overhead for module importance ranking on the intra-domain server side. Assuming the number of selected clients in the same domain is k, and each client includes {32d} LoRA (64 modules), the sorting time complexity is k \times 64 \times \log(64). In the financial domain, k=5; in the general domain, k=2; and in the medical domain, k=2. The entire experiment ran on an NVIDIA A40 GPU for five and a half hours.
>
> *Convergence Results*: As shown in Figure 1 of the PDF, our DoFIT demonstrates faster and more stable convergence compared to FIT using FedAvg and FedProx as the FL framework in both single-domain and dual-domain scenarios.
>
> *Client Numbers*: 50(5) \& 20(2) indicate that in the financial domain, there are 50 clients in total, with 5 clients randomly selected for training and uploading each round. In the general domain, there are 20 clients in total, with 2 clients randomly selected for training and uploading each round. As shown in Table 3 of the PDF, varying the total number of clients or the number of selected clients does not cause significant fluctuations in the results, demonstrating that the proposed DoFIT is robust to the number of clients.
>
> ---
>
> **5:** Thanks. The importance of modules in LoRA varies across different domains, indirectly reflecting domain heterogeneity. As shown in Figure 2 (left) of the PDF, the top-15 important modules in domains F and G are not completely the same. As training progresses, the weights of the same modules become more reinforced.
>
> We also further compute the L2 norm of the difference in the singular value spectrum between each client's LoRA and the global LoRA for the same domain and different domains. As shown in Figure 2 (right) of the PDF, this visualization reflects smaller intra-domain data heterogeneity and greater inter-domain data heterogeneity.
>
> > Questions:
>
> 1,2,3: See 1,3,4.

---

### Official Review · Reviewer_xCSC · 2024-07-06

**Soundness:** 4
**Presentation:** 3
**Contribution:** 3
**Rating:** 7
**Confidence:** 5

**Summary:**

This work offers a solution to a problem in the collaborative training of different domains on decentralized data within the FIT paradigm: domain-aware data heterogeneity causes domain-information catastrophic forgetting. The solution relies on two new designs for aggregation and initialization. Specifically, in the aggregation step, DoFIT-base aggregates overlapping inter-domain information at a fine granularity on the inter-domain server side. In the initialization step, DoFIT projects modules with inter-domain information onto parameter regions least affected by intra-domain update. Finally, the authors conducted extensive comparison experiments to well show the significant effectiveness of the proposed method.

**Strengths:**

1. The authors introduce a novel FIT framework aimed at solving the domain-information catastrophic forgetting problem.
2. Considering that existing FIT methods struggle with the variation from different domains, resulting in inferior results for the original specific domain, the proposed DoFIT outperforms conventional FIT methods by aggregating overlapping inter-domain information and initializing with inter-domain information.

**Weaknesses:**

1. Although the current framework performs better than conventional frameworks in F&G or M&G domains, can it be further applied to more domains?
2. There are related articles that test the FL results of different domain data in different clients, where each client has data from one domain data. Thus, it is not clear how the cross-domain training problem differs in this paper.

**Questions:**

1. In the DoFIT framework, the final test determines whether to experiment with global LoRA on the inter-domain server side or the intra-domain server side.
2. Is it more reasonable that the scaling factors of LoRA B and A are not consistent?
3. There are too many symbols in the methods section; it is best to use a table to clearly specify the important symbolic variables.
4. The difference between DoFIT-base and Conventional FIT in Section 3.2 is best highlighted.

**Limitations:**

no limitation in this scope.

---

> ### Author Rebuttal · Authors · 2024-08-06
>
> > Weaknesses:
>
> 1: Thank you. Currently, the experiments show improvements only between two domains. Due to the more complex and variable heterogeneity in multi-domain scenarios, the current framework cannot handle it well. However, as the first federated instruction tuning framework attempting to address domain-aware data heterogeneity, it provides some inspiration for future FIT methods dealing with more domains.
>
> 2: Thank you. The related work on FIT tested FL results with one client per domain, which only addressed inter-domain data heterogeneity. However, our DoFIT accounts for multiple clients within a domain, addressing both intra-domain and inter-domain data heterogeneity issues.
>
> > Questions:
>
> 1: The final test involves first merging the LLMs with global LoRA on the intra-domain server side, and then testing on the current domain dataset.
>
> 2: The scaling factors for LoRA B and A are consistent and already yield good results. Using different scaling factors would increase the burden of hyperparameter tuning and make it more challenging for the model to generalize across different domains.
>
> 3: Thank you. We will add descriptions of the important symbolic variables in the final version.
>
> 4: Thank you. The first and second paragraphs of Section 3.2 in the original paper explain the differences between DoFIT-base and conventional FIT.

---

### Official Review · Reviewer_ZeZJ · 2024-07-06

**Soundness:** 3
**Presentation:** 3
**Contribution:** 3
**Rating:** 8
**Confidence:** 4

**Summary:**

The authors propose to utilize intra- and inter-domain server sides in a federated instruction tuning framework to implement discriminative aggregation and initialization strategies. The proposed approach, i.e., DoFIT, is primarily based on FIT of LLM, marking the first solution to address domain-aware data heterogeneity in collaborative training on decentralized data for the FIT paradigm. Unlike conventional FIT, which ignores the variation between data from different domains, DoFIT enables the intra-domain server to perform normal aggregation and initialization, while the inter-domain server handles overlapping weights aggregation and less-conflicted initialization. Consequently, DoFIT reduces interference information and preserves more domain information to mitigate catastrophic forgetting. The authors conducted empirical comparisons with conventional FIT methods on three datasets from finance, general, and medical domains.

**Strengths:**

A novel and promising application involves supplementing data from other related domains is explored to develop a powerful model when data within a specific domain is scare.
A new domain-aware FIT framework that includes fine-grained inter-domain aggregation is proposed to address domain-aware data heterogeneity.
A novel initialization strategy in intra-domain global LoRA is presented to alleviate catastrophic forgetting.

**Weaknesses:**

1. In the F&G and M&G domains, the values of k and α in DoFIT are quite different. Could adding more hyperparameters potentially harm the generalization of DoFIT?
2. There are fewer methods compared in the single-domain experiments, and it seems that more comparisons of FL methods other than FedAvg should be added.
3. The proposed intra- and inter-domain servers in DOFIT-Base and DoFIT increase the communication cost compared to traditional FIT methods.

**Questions:**

1. The meaning of “Overlap” in Figure 1 is unclear and should be explained, preferably in the title.
2. The set of modules where “Overlap” resides is in units of LoRA B or A, while the updating weight matrix is in units of each layer in the transformer architecture and should be indicated.
3. Is the scaling factor α in Eq. 10 the same for LoRA B and A?

**Limitations:**

No limitation is explained in the paper.

---

> ### Author Rebuttal · Authors · 2024-08-06
>
> > Weaknesses:
>
> 1: In our DoFIT, we only added two hyperparameters: top-k and α. As shown in Tables 1 and 2, as well as Figures 4 and 5 of the original paper, the performance differences for different top-k and α values are minimal. Therefore, even when generalizing to new domain data, extensive adjustment of these two hyperparameters is unnecessary. The proposed DoFIT demonstrates a certain robustness to these hyperparameters.
>
> 2: Thank you. We think that incorporating traditional FL methods only addresses client-aware data heterogeneity within the same domain and does not tackle domain-aware data heterogeneity. Therefore, we only compared a few representative single-domain FL methods in the paper. As suggested, we included more comparison methods (FedProx [1]), and the results further demonstrated the effectiveness of our approach, as shown in Figure 1 of PDF.
>
> [1] Li, Tian, et al. "Federated optimization in heterogeneous networks." Proceedings of Machine learning and systems (MLsys), 2020, 2: 429-450.
>
> 3: Thank you. In Table 3 of the original paper, the communication cost (S-Comm.) added by DoFIT-Base and DoFIT compared to FIT is shown for intra-domain and inter-domain servers. Compared to FIT, DoFIT introduces only a slight increase in communication parameters between intra- and inter-domain server sides (indicated by S-Comm.), with minimal impact on well-resourced servers (also indicated by S-Comm.).
>
> > Questions:
>
> 1: Thank you. As mentioned on line 164 of the original paper, "overlapping" refers to both the same layer and the same decomposition components. We will include this explanation in the final version of Figure 1’s title.
>
> 2: Thanks for the suggestion. We will clarify this in the revised version.
>
> 3: Yes, the scaling factor α in Eq. 10 is the same for LoRA B and A.

---

> > ### Comment · Reviewer_ZeZJ · 2024-08-09
> >
> > Thank you for taking the time to write a detailed response! It has definitely improved my understanding of the work and helped me appreciate its importance.
> > The authors' response seems to resolve most of my concerns. Ultimately, this paper investigates interesting new approaches in utilizing intra- and inter-domain servers and proposes a simple solution. Accordingly, I will update my score from 6 (wa) to 8 (sa).

---

### Official Review · Reviewer_Vje9 · 2024-07-08

**Soundness:** 3
**Presentation:** 2
**Contribution:** 3
**Rating:** 6
**Confidence:** 4

**Summary:**

In this work, the issue of domain-aware data heterogeneity is solved, which equally treats intra- and inter-domain data variations but cannot adapt to the challenges of  cross-domain training. The paper proposes a novel domain-aware federated instruction tuning (DoFIT) framework for collaborative training across datasets in related domains to enhance the performance of individual domains. Specifically, DoFIT aggregates the top-k important modules and initializes intra-domain modules with addition of a proximal perturbation term. The proposed DoFIT demonstrates a significant performance improvement when transitioning from single-domain to dual-domain datasets.

**Strengths:**

1.The proposed DoFIT is novel and interesting. Instead of treating intra- and inter-domain data heterogeneity equally, the authors aggregate the top-k important modules on the inter-domain server side and initialize intra-domain modules with the addition of a proximal perturbation term on the intra-domain server side. These targeted processing schemes alleviate the catastrophic forgetting problem.
2.The motivation and techniques of this paper are reasonable. In DoFIT, the aggregation of important modules is motivated by the fact that the larger LoRA weight has the greater impact on the frozen LLM. In DoFIT, proximal perturbation initialization is derived from the orthogonal learning approach.
3.The comparison of communication costs is detailed in Table 3.
4.The provided experiment results are convincing.

**Weaknesses:**

1.The concept of different domains is less clearly defined, and the scope of application of the proposed framework seems limited.
2.The experiments only yield results on single-domain and dual-domain datasets, and cannot be extended to multi-domain scenarios.
3.The privacy and security of the intra- and inter-domain server sides do not seem to be addressed.

**Questions:**

Whether FIT32qv differs between single-domain and dual-domain scenarios, and how to extend it to dual domains should be explained.

**Limitations:**

The authors provide limitations and future works.

---

> ### Author Rebuttal · Authors · 2024-08-06
>
> > Weaknesses:
>
> 1: Thanks. Unlike traditional FL methods that focus on different styles within the same category, the different domains in this paper refer to various application scenarios, such as the financial, general, and medical domains. The proposed framework is suitable for federated instruction tuning tasks across different but related domains.
>
> 2: Thanks. As the first attempt to address domain-aware data heterogeneity in federated instruction tuning, our proposed framework is experimentally demonstrated the effectiveness on both single and dual domain datasets. Extending to multi-domain scenarios is a promising future work and we believe our work could lay a foundation and provide insights for deeper exploration in this area.
>
> 3: Thank you. The privacy and security of intra- and inter-domain servers are the same as in traditional FIT methods with a single server. Compared to previous FIT methods, we do not upload more parameters or information to the servers.
>
> > Questions:
>
> FIT32qv extends from the financial domain to both financial and general domains by simply adding clients from the general domain. It still uses a single server to equally aggregate parameter weights from clients in both domains.

---

### Author Rebuttal · Authors · 2024-08-06

> We thank the reviewers for their valuable comments. We are glad that the reviewers found:

- The topic we are addressing to be promising and important (Reviewers ZeZJ, Zbxm, DeWw).
- Our experiments to be convincing and showing significant performance improvement (Reviewer Vje9, DeWw).
- Our idea of aggregating and initializing to be novel and compelling (Reviewers Vje9, ZeZJ, xCSC, DeWw).
    - However, there may be a conflict with the novlty issue concerned by Reviewer Zbxm.

> We have responded to the comments given by the reviewers carefully. Here we summarize a few important points of our rebuttal or revision.

1.We explained the differences between federated domain adaptation for LLMs and existing federated domain adaptation methods.

2.Our experiments include:

* Comparison with the existing federated domain adaptation method FedGP.
* New criteria: gradient and singular value.
* Complexity, convergence, and client numbers.
* Visualization of domain heterogeneity.

3.Federated domain adaptation for LLMs is crucial, yet no methods exist. Our work is the first FIT framework to address domain-aware data heterogeneity, offering inspiration for future research.

---

### Decision · Program_Chairs · 2024-09-25

**Decision:**

Accept (poster)

**Comment:**

Most reviewers agree that the proposed approach DoFIT is a novel and insightful approach to handle intra-domain and inter-domain data heterogeneity in federated instruction tuning for LLMs.
In particular, DoFIT performs inter-domain aggregation, i.e., domain-aware updates, by aggregating overlapping modules between domains.
To avoid catastrophic forgetting, DoFIT projects inter-domain information onto parameter regions that are less affected by intro-domain updates.
The method demonstrates that aggregating across two domains (e.g., finance and general) can enable boost in performance.
Furthermore, the experiments on finance, general, and medical domains give evidence for the usefulness of DoFIT, outperforming other methods.

While there was some concern about the method's novelty with respect to federated domain adaptation, which also handles domain heterogeneity, the authors explained that LLM-based FIT has unique characteristics that distinguish it from the standarnd non-LLM setting and provided evidence that prior FL work does not fill this gap.
Other minor concerns included the restriction to only two domains rather than multi-domain. However, the strong results on two domains, which could be a realistic situation, are sufficient to demonstrate the effectiveness of the method.
The authors should include their new results and discussion on these points in their final version.

Overall, the paper presents an effective approach to federated instruction tuning for LLMs that appropriately handles domain heterogeneity such that it avoids catastrophic forgetting and shows improved performance.